# Strategies to Control In Vitro Degradation of Mg Scaffolds Processed by Powder Metallurgy

Sandra C. Cifuentes [1,*], Lucía Alvarez [2], Luis Arias [2], Tobias Fey [3,4] and Sophia A. Tsipas [2]

1   Department of Applied Mathematics, Materials Science and Engineering and Electronic Technology, Escuela Superior de Ciencias Experimentales y Tecnología (ESCET), Universidad Rey Juan Carlos, 28933 Madrid, Spain
2   Grupo de Tecnología de Polvos, Departamento de Ciencia e Ingeniería de Materiales e Ingeniería Química, IAAB, Universidad Carlos III de Madrid, 28911 Madrid, Spain; lucia.alvarez@outlook.es (L.A.); luandar2592@gmail.com (L.A.); stsipas@ing.uc3m.es (S.A.T.)
3   Department of Materials Science and Engineering (Glass and Ceramics), Friedrich-Alexander-Universität Erlangen-Nürnberg, Martensstr. 5, D-91058 Erlangen, Germany; tobias.fey@fau.de
4   Frontier Research Institute for Materials Science, Nagoya Institute of Technology, Gokiso-cho, Showa-ku, Nagoya 466-8555, Japan
*   Correspondence: sandra.cifuentes@urjc.es

**Abstract:** Magnesium scaffolds are biodegradable, biocompatible, bioactive porous scaffolds, which find applications within tissue engineering. The presence of porosity increases surface area and enhances cell proliferation and tissue ingrowth. These characteristics make Mg scaffolds key materials to enhance the healing processes of tissues such as cartilage and bone. However, along with the increment of porosity, the corrosion of magnesium within a physiological environment occurs faster. It is, therefore, necessary to control the degradation rate of Mg scaffolds in order to maintain their mechanical properties during the healing process. Several studies have been performed to increase Mg scaffolds' corrosion resistance. The different approaches include the modification of the Mg surface by conversion coatings or deposited coatings. The nature of the coatings varies from ceramics such as hydroxyapatite and calcium phosphates to polymers such as polycaprolactone or gelatin. In this work, we propose a novel approach to generating a protective bilayer coating on the Mg scaffold surface composed of a first layer of naturally occurring Mg corrosion products (hydroxide and phosphates) and a second layer of a homogeneous and biocompatible coating of polylactic acid. The Mg scaffolds were fabricated from Mg powder by means of powder metallurgy using ammonium bicarbonate as a space holder. The size and amount of porosity were controlled using different size distributions of space holders. We addressed the influence of scaffold pore size on the conversion and deposition processes and how the coating process influences the in vitro degradation of the scaffolds.

**Keywords:** Mg scaffolds; biodegradable; powder metallurgy; in vitro degradation

## 1. Introduction

The development of synthetic scaffolds seeks to provide an alternative to autografts, allografts, and xenografts for the regeneration of bone tissue defects. The objective is to overcome the drawbacks of biological materials such as availability, donor site morbidity (autografts), and the risk of infection and disease transmission (allografts and xenografts). Commercially available synthetic scaffolds based on calcium phosphates in the form of hydroxyapatite or β-tricalcium phosphate (β-TCP) are nowadays a clinical solution for contained defects [1,2]. These materials have osteoconductive, osteointegrative, and osteoinductive capabilities but they exhibit a fragile mechanical performance and they lack antimicrobial properties; therefore, they are not a solution for large bone defects [3,4]. Extensive research has been performed on biodegradable scaffolds based on FDA-approved polymers such as polycaprolactone (PCL), polylactic acid (PLA), and poly(lactic-co-glycolic) acid (PLGA) [5].



These materials have very interesting viscoelastic properties, are biodegradable, and their scaffolding technologies are versatile. However, they are not mechanically stable enough and present other issues as their lack of bioactivity and long resorption times, together with the degradation by-products, lead to inflammation, and their intrinsic hydrophobic properties limit their biological performance [6,7].

In this context, magnesium appears as a promising material in tissue engineering with the potential to become a real alternative to biological scaffolds and a solution for critical size defects. Metallic Mg has mechanical properties comparable to bone tissue and is bioactive and bioresorbable. Mg ions are significant for bone growth as they promote osteoblast proliferation and differentiation [8]. Mg also has shown antibacterial properties [9]. However, this metal has a low corrosion resistance, which leads to three main adverse effects: (1) the release of hydrogen gas that forms subcutaneous bubbles, (2) the early loss of mechanical integrity prior to bone healing, and (3) local alkalization that compromises the equilibrium of physiological reactions that depend on pH [8]. It is, therefore, necessary to control the degradation rate of Mg in accordance with the bone healing process to meet the requirements for tissue engineering.

Extensive studies have been performed to increase the corrosion resistance of non-porous bulk Mg. Surface modifications have been demonstrated to be effective, and Hornberger et al. comprehensively reviewed the strategies to control magnesium alloys' degradation through coatings [10]. The different approaches have been thoroughly studied for non-porous bulk Mg alloys, but in a lesser extent for Mg scaffolds.

Scaffolds' porosity is a crucial morphological property to enhance cell proliferation and tissue ingrowth. Interconnected porous networks are preferred rather than closed pores. Open porosity is osteoconductive and it allows cell migration and vascularization [11]. Pore sizes for optimum cell proliferation and differentiation are recommended to be between 150 and 500 µm [12].

However, along with the increment in porosity, the surface area also increases and, therefore, the corrosion of scaffolds occurs faster than in non-porous bulk materials and is more challenging to control. According to Yazdimamaghani et al., the most common strategies used for increasing Mg scaffolds' corrosion resistance are chemical conversion layers and deposited inorganic or organic coatings [13]. In this sense, porous magnesium structures have been coated with hydroxyapatite [14], β-tricalcium phosphate [15,16], magnesium fluoride [17–19], bioglass [20], or polycaprolactone [21]. Although a single coating manages to improve the corrosion resistance of magnesium foams considerably, several in vivo studies have demonstrated that one single coating is not enough to modulate the corrosion rate of these scaffolds; their corrosion rate is still too fast for osseointegration to take place [17,18,22–24]. Therefore, there is a need for new strategies to further slow the degradation rate of Mg scaffolds.

Some authors have proposed multilayer coatings to enhance the corrosion resistance of porous magnesium. Table 1 collects the works that have been carried out in the field of bilayer coatings on porous Mg. For instance, Yazdimamaghani et al. developed a coating composed of polycaprolactone and gelatin reinforced with bioactive glass on Mg scaffolds produced by powder metallurgy [25]. Julmi et al. [26] and Maier et al. [27] coated investment-casted Mg scaffolds (LAE442, Mg-La2 and ZX61) with bilayer coatings based on a bottom layer of $MgF_2$ and a top layer of polylactic acid (PLA) or calcium phosphate. Zhang et al. [23] modified the surface of die-casted MgZnCa scaffolds with a microarc oxidation (MAO) bottom layer and a top layer composed of calcium phosphates.

In this work, we developed a bilayer biodegradable coating on Mg scaffolds through an approach that has not been tried before. We proposed a coating composed of a first layer containing magnesium hydroxide and magnesium phosphates developed by chemical conversion and a second layer of a deposited organic coating of polylactic acid. Mg hydroxide and phosphates are biodegradable and non-toxic. $Mg(OH)_2$ is the main corrosion product of Mg alloys and the major origin of the observed enhanced bone growth in vivo of these alloys [28]. Mg phosphates promote osteoblast differentiation and stimulate

osteogenesis similarly to calcium phosphates [29]. Polylactic acid is an FDA-approved resorbable polymer, which has been demonstrated to increase the corrosion resistance of Mg alloys when applied as a coating [22,30].

**Table 1.** Bilayer coatings on Mg scaffolds.

| Mg Alloy | Scaffold Fabrication Process | Porosity/Pore Size | Chemical Conversion Layer | Deposition Layer | Reference |
|---|---|---|---|---|---|
| Mg | Powder Metallurgy Space holder | 35–40%/150–300 µm | | PCL-BaG + Gel-BaG | M. Yazdimamaghani et al. [25] |
| MgZnCa | Die casting | Drilled Holes (1-mm diameter) | Microarc oxidation + Calcium phosphates | | N. Zhang et al. [23] |
| LAE442 and Mg-La2 | Investment casting | >40%/400–500 µm | $MgF_2$ | PLA | S. Julmi et al. [26] |
| | | | $MgF_2$ + Calcium Phosphate | | |
| Mg | Space holder method (NaCl) and spark plasma sintering | 30–70%/200–400 µm | Hydroxyapatite | Hybrid poly (ether imide) (PEI)-$SiO_2$ | M.-H. Kang et al. [31] |
| Mg-La2, LAE442, ZX61 | Investment casting | >40%/500 µm | $MgF_2$ | PLA | H.J. Maier et al. [27] |
| | | | $MgF_2$ + Calcium Phosphate | | |

PCL: polycaprolactone; BaG: bioactive glasses; Gel: gelatin.

We fabricated scaffolds through the powder metallurgy route using the space holder method given its cost effectiveness and ease of processing open-porous interconnected structures over other techniques such as negative salt pattern or titanium wire space holder [13,32]. Most of the studies on Mg scaffolds for biomedical applications focus mainly on the influence of the fabrication process on the porous structure and the effect of that porosity on the mechanical properties and the degradation rate [32,33]. In this research, we addressed the effect of the pore size on the coating process and how the coating process influences the in vitro degradation behavior of the grafts. We compared scaffolds with two different pore size ranges (250–400 µm and 400–800 µm) and analyzed the effect of these pore sizes on the chemical conversion process and the deposition of the polymeric layers. Additionally, we compared two methods for the deposition of the top polymeric layer (dip-coating and vacuum-assisted infiltration). We evaluated the in vitro degradation of uncoated porous Mg, scaffolds coated with PLA, and scaffolds with the bilayer coating by assessing the hydrogen release and pH evolution.

## 2. Materials and Methods

### 2.1. Starting Powders

Mg scaffolds were fabricated using irregular Mg flake-like particles (Goodfellow) (Figure 1a) with a purity of 99.9% and a length-to-width ratio of 1.6:1. The average particle size ($D_{10}$: 13.1 µm, $D_{50}$: 25.9 µm, $D_{90}$: 46.9 µm) was determined using a Mastersizer 2000 analyzer (Malvern Instruments, Malvern, UK) (Figure 1c). Ammonium bicarbonate (Sigma-Aldrich, San Luis, MI, USA) was used as a space holder, with a particle size ranging from 10 to 800 µm (Figure 1b). The powder was sieved, and four size distributions were obtained: 10–125 µm, 125–250 µm, 250–400 µm (Figure 1d), and 400–800 µm.

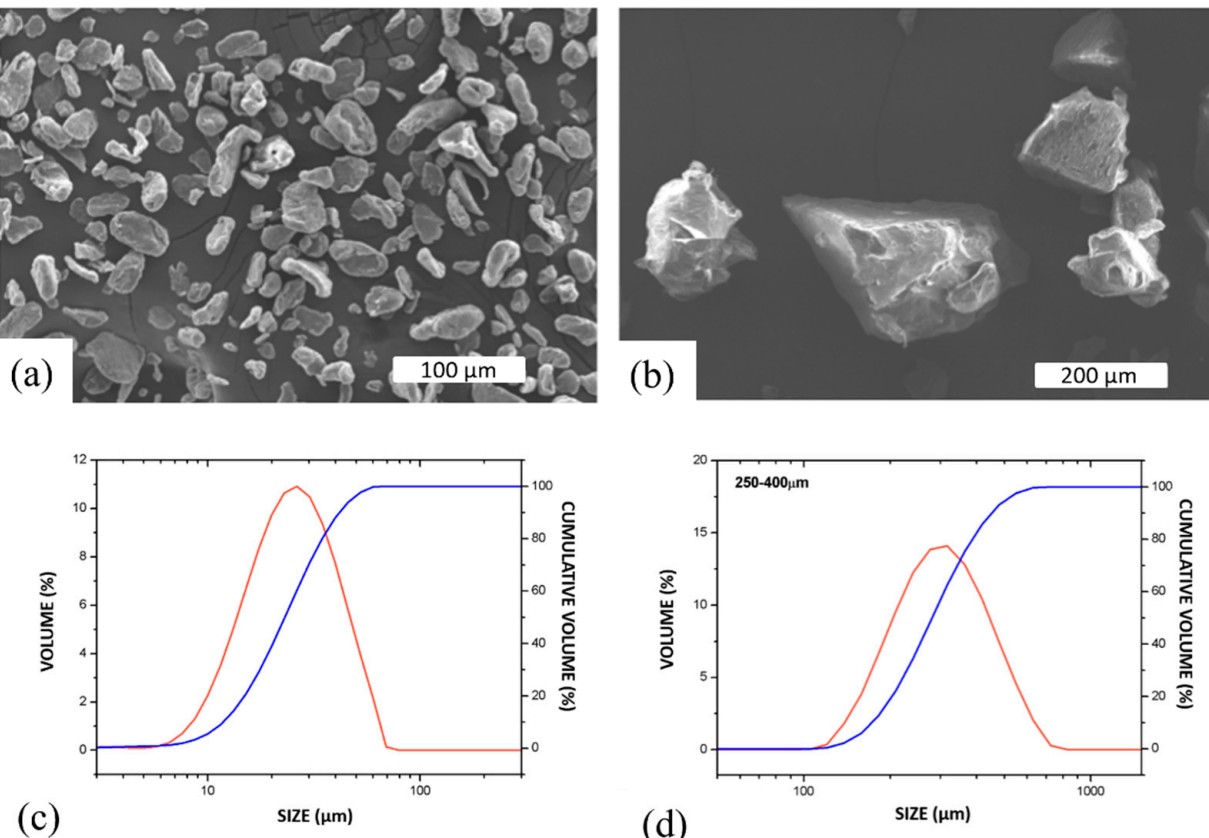

**Figure 1.** Characteristics of starting powders. Scanning electron microscope (SEM) images of (**a**) Mg and (**b**) $NH_4HCO_3$ powders. Particle size distribution of (**c**) Mg and (**d**) $NH_4HCO_3$ powders measured in water.

### 2.2. Scaffolds' Fabrication

A mixture of Mg/$NH_4HCO_3$ with a volume ratio of 60:40 was blended in a Turbula mixer for 30 min (WAB Group, Muttenz, Switzerland). Mixtures were then uniaxially pressed into disc compacts (diameter: 16 mm, height: 4.5 mm) at a pressure of 300 MPa. The space holder was decomposed by heating the samples in a furnace at 140 °C for 2 h under argon atmosphere. Samples were sintered at 580 °C for 2 h under two different atmospheres: (1) argon atmosphere using Mg powder as an oxygen getter and (2) low vacuum at a pressure of $2.9 \times 10^{-2}$ mbar. The mass of the specimens was measured before and after sintering using a balance (Sartorius, Göttingen, Germany) with a precision of $\pm 0.0001$ gr.

### 2.3. Scaffolds' Characterization

The overall, open, and closed porosity of the scaffolds were calculated by Archimedes' method following the ASTM C20-00 standard and using ethanol to infiltrate the samples and as an immersion medium. Longitudinal cross-sections were metallographically prepared to examine the microstructure of the scaffolds by SEM-EDX (Philips XL 30, SEMTech Solutions, Billerica, MA, USA). X-ray diffraction technique (Philips X'Pert X-ray diffractometer, Malvern Panalytical, Malvern, UK) was used to assess the oxygen uptake of Mg during sintering under both atmospheres (argon and low vacuum).

X-ray microtomography (μCT) was used to obtain 3D volume data from the samples using a Skyscan 1172 with a 11 MP CCD camera (Skyscan, Konitch, Belgium) equipped with a tungsten tube (λ = 0.024 nm). The measurements were performed at 80 kV and 100 μA using a combined Al+Cu filter to reduce low-energy noise radiation and a rotation of 180 °C with a rotation step of 0.7. The resulting resolution was 23.77 μm per voxel. Reconstruction

of the sinograms was done by NRecon Software (V. 1.7.4.2, Bruker, Billerica, MA, USA) to create a set of 2D image slices. Amira 2021.2 (Thermo Fisher Scientific Inc., Waltham, MA, USA) was used for segmentation, visualization, and pore network analysis. Details on the pore network analysis can be found in [34]. The cell and strut size distributions were calculated by CTAnalyzer (Skyscan, Kontich, Belgium). The scaffolds studied by this technique were those processed using ammonium bicarbonate with the following size distributions: 250–400 μm and 400–800 μm.

### 2.4. Development of Coatings on Scaffolds

Open-porous scaffolds were evaluated in three conditions: (1) uncoated scaffolds (Mg), (2) scaffolds with a polylactic acid (PLA) coating (Mg + PLA), and (3) scaffolds treated by immersion in phosphate-buffered saline solution (PBS) prior to deposition of PLA (Mg + PBS + PLA) (Figure 2). Semi-crystalline polylactic acid (PLA) was used for coating the scaffolds. Nature Works LLC (Minneapolis, MI, USA) supplied the polymer under the commercial name of PLA Polymer 7032D. The PLA had a specific gravity of 1.24 and a Melt Flow Index (210 °C g/10 min) between 5 and 15. PLA was dissolved in chloroform at different concentrations (3 g/100 mL, 6 g/100 mL, and 12 g/100 mL). Polymeric coatings were applied by two different methods: dip-coating and a vacuum-assisted infiltration technique. The scaffolds' surface was treated by immersion of the specimens in phosphate-buffered saline (PBS) solution (Thermo Fisher Scientific, Waltham, MA, USA) for 30, 60, and 120 min. The microstructure and composition of the resulting layer was analyzed by SEM-EDX (Phillips XL 30, SEMTech Solutions, Billerica, MA, USA).

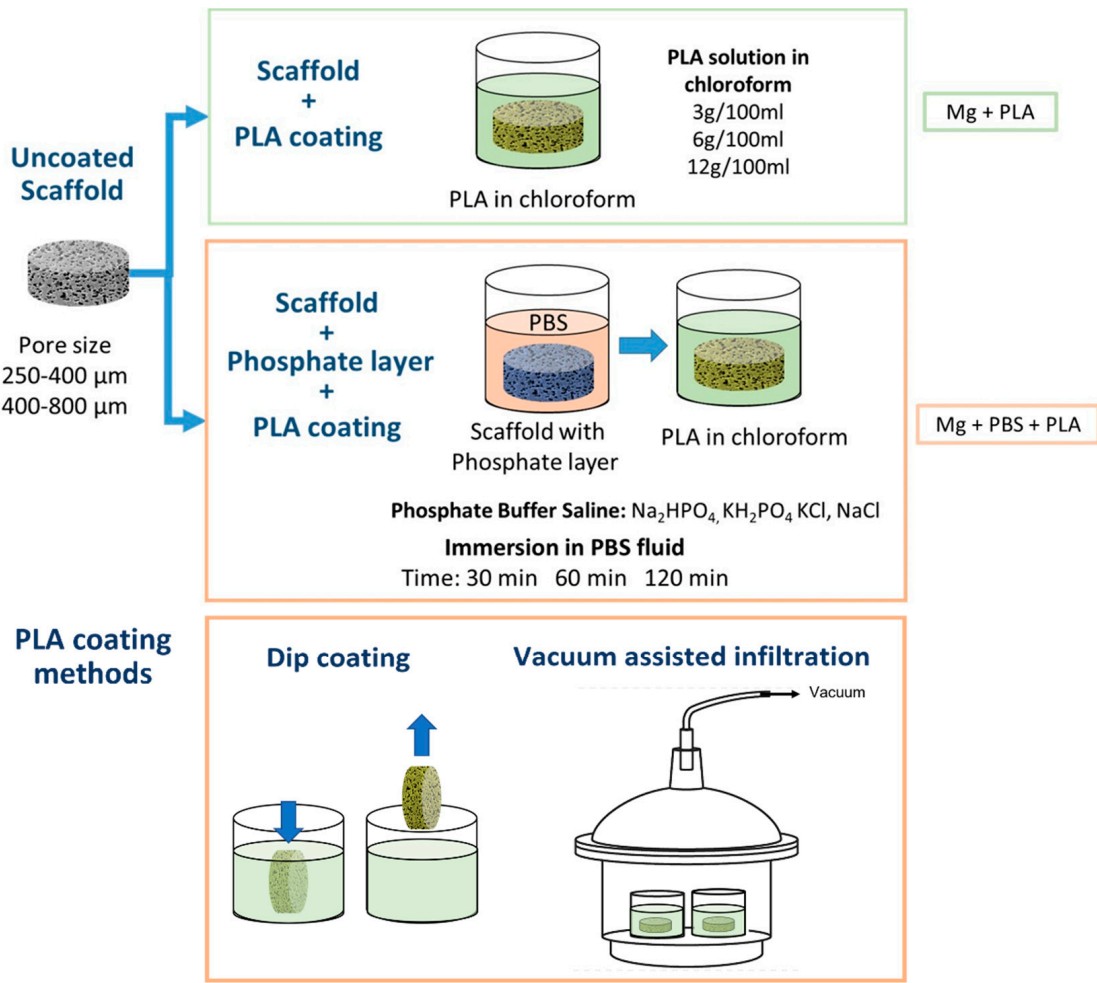

**Figure 2.** Schematic diagram of surface treatments applied to Mg scaffolds.

*2.5. In Vitro Degradation Assessment by pH Evolution and Hydrogen Release*

Mg scaffolds, PLA-coated scaffolds, and Mg + PBS + PLA scaffolds were immersed in PBS at 37 °C and the evolution of pH was monitored for 2 hours (Thermo Scientific™ Orion™ 720A+, Waltham, MA, USA). Measurement of hydrogen release was performed also in PBS at 37 °C, using 35 mL of solution for each sample. Three specimens were used for each condition. Hydrogen release was measured using a setup consisting of a beaker, a glass funnel, and a burette. A sample was placed inside the beaker, covered with the glass funnel, and placed upside down. The burette was placed inverted over the funnel so it could collect the hydrogen gas released during Mg degradation. Experiments were performed for 5 days. The results correspond to the average values of three specimens.

## 3. Results and Discussion

Figure 3a shows the difference in mass (mass gain or loss) of sintered samples as a function of space holder particle size for each sintering atmosphere (argon and vacuum). It was observed that samples sintered under argon atmosphere gained mass, between 1.2% and 1.5%. This small increment in mass could be attributed to the oxidation of Mg particles due to the small oxygen content as argon impurity. Samples sintered in vacuum conditions showed mass loss. The finer the space holder size was, the higher the mass loss was. Scaffolds fabricated with a 400–800-µm space holder lost 10% of mass, and scaffolds fabricated with a 10–125-µm space holder lost 18% of mass. The mass loss could correspond to a sublimation of Mg given the low-pressure conditions. The increment in mass loss with smaller space holder particle size could be explained in terms of the increment of the surface area for scaffolds with smaller pores (i.e., fabricated with finer ammonium bicarbonate particles). Considering the mass change results at both sintering conditions, argon and vacuum, the study proceeded with the scaffolds fabricated under argon atmosphere to avoid sublimation of Mg during processing.

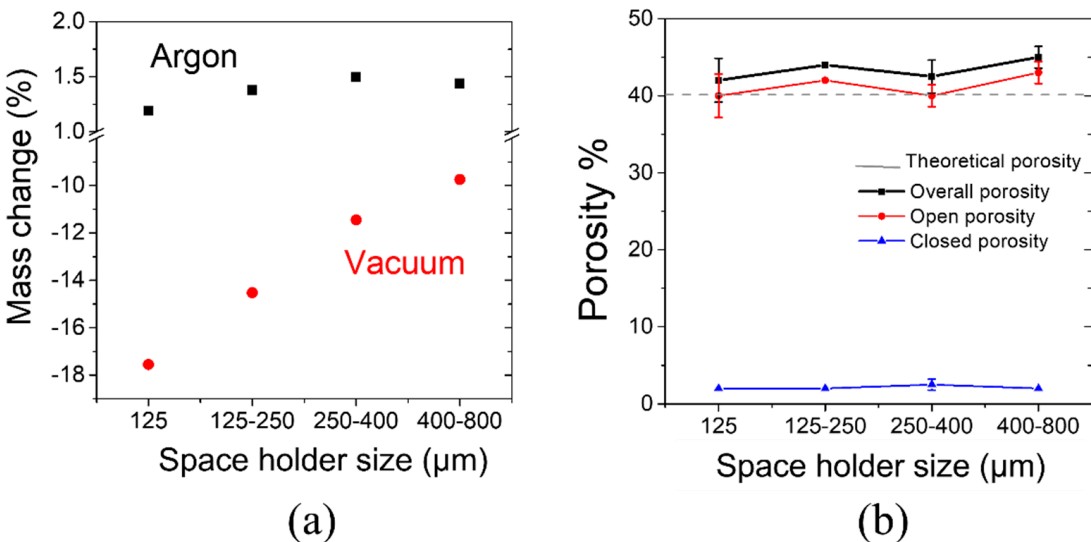

**Figure 3.** (**a**) Weight change of scaffolds produced with different pore sizes, after sintering in different atmospheres; (**b**) total, open, and closed porosity of scaffolds with different pore sizes sintered under argon atmosphere.

The results of total, open, and closed porosity of scaffolds sintered in argon atmosphere are shown in Figure 3b and were compared to the theoretical porosity. The theoretical porosity refers to the volume percentage of a space holder added to Mg powder during scaffolds' fabrication (40%). Magnesium scaffolds exhibit mainly open porosity, which is necessary for their applications as bone grafts. Closed porosity was only about 2% for all space holder sizes and it was attributed to porosity present in the cell wall. Complete

elimination of the space holder was ensured and monitored by weighing the samples before and after the space holder decomposition heat treatment. For all cases, it was confirmed that 100% of the mass of the space holder was eliminated during decomposition. Magnesium scaffolds exhibited mainly open porosity, with a homogeneous distribution of pores and controlled pore size for all space holder size distributions. The microstructures of scaffolds are shown in Figure 4a–d.

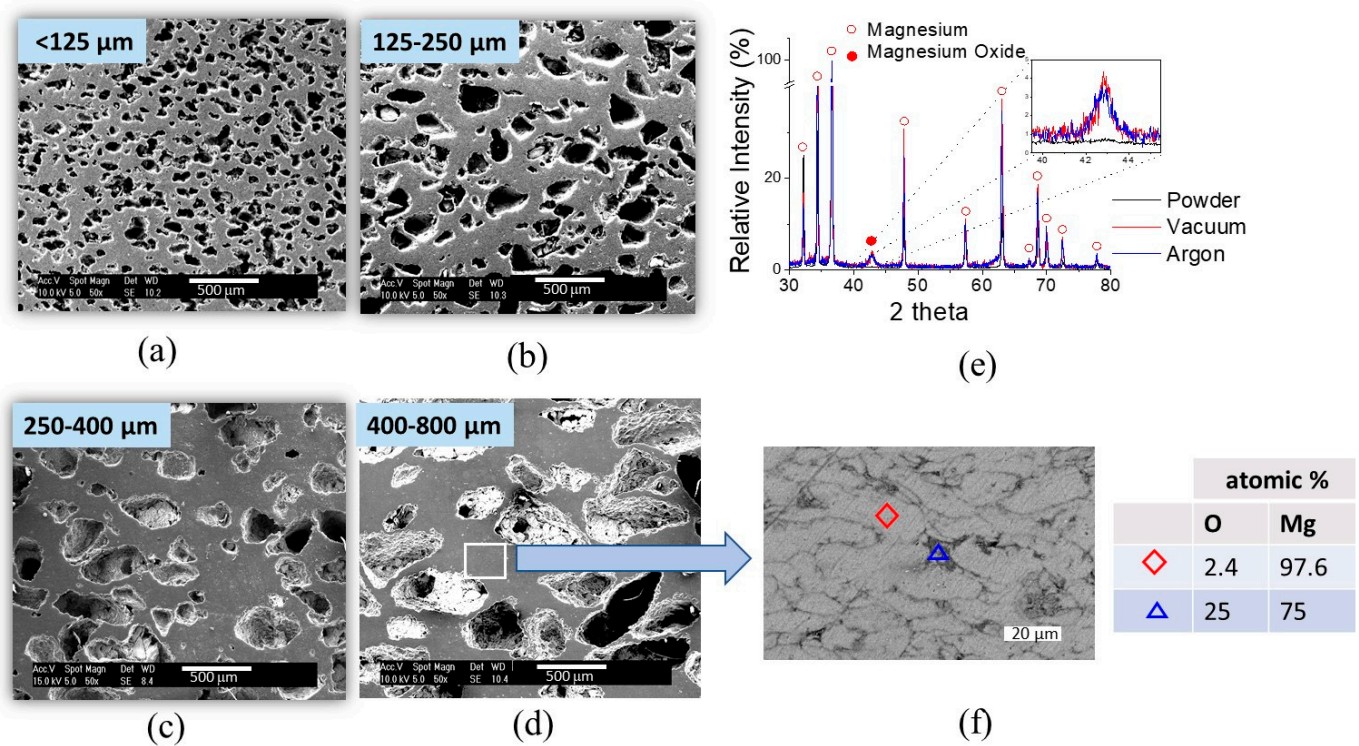

**Figure 4.** SEM micrographs of Mg scaffolds sintered under argon atmosphere and produced with a 40 vol.% of space holder with a particle size of: (**a**) 10–125 μm; (**b**) 125–250 μm; (**c**) 250–400 μm; and (**d**) 400–800 μm; (**e**) X-ray diffraction pattern of Mg scaffold; (**f**) SEM micrograph (BSE detector) of detail in cell neck in a Mg scaffold present (light gray: Mg particle; dark gray: particle boundaries) and table with EDX analysis in Mg particle and particle boundaries.

Figure 4f shows a backscattered electron image of the cell neck between pores in the scaffold and the results of the EDX analysis on the detail of the Mg scaffold cell strut. The image shows the presence of oxide (dark gray) surrounding Mg particles (light gray).

Figure 4e shows the X-ray diffraction patterns of the scaffolds sintered under Argon atmosphere and under vacuum compared with the XRD pattern of Mg starting powder. The three main characteristic reflections of Mg at 32.2°, 34.4°, and 36.6°, which correspond to the (100), (002), and (101) planes, respectively, can be clearly distinguished. Magnesium oxide presented a main peak at 43°. The diffractograms suggest that there was no significant difference in the amount of MgO present in the scaffolds sintered under argon atmosphere and vacuum conditions. The intensity of the peak corresponding to MgO was very similar for both sintering conditions. However, diffractograms of the Mg starting powder did not exhibit peaks that could indicate the presence of MgO, which suggests that the oxide was formed during the scaffolds' fabrication. It is possible that MgO might form during the decomposition of the $NH_4HCO_3$ pore former. Nevertheless, the presence of a small amount of MgO in the scaffold was not considered to be of significance for the performance of the scaffold. The formation of small amounts of MgO is very difficult to avoid when processing Mg [35].

Some studies of the production of Mg scaffold by powder metallurgy methods use similar sintering conditions. Yazdimamaghani et al. fabricated Mg scaffolds using carbonate

hydrogen ammonium as a space holder. They performed a heat treatment process at 175 °C for 2 h to evaporate the space holder and at 600 °C for 2 h for sintering Mg particles [25]. Zhuang et al. used carbamide particles as a space holder and the heat treatment process consisted of the evaporation step at 250 °C for 4 h and sintering at 500 °C for 2 h [36]. Other studies that focused on dense non-porous Mg usually employed much longer sintering times to achieve complete densification of Mg particles [35]. The sintering conditions used here (2 h at 580 °C in argon atmosphere) enabled the production of scaffolds with an open porosity very close to the theoretical one (40%) and a closed porosity around 2%. Closed porosity is usually related to the later sintering stages; it is difficult to further eliminate and reach higher densifications. Hence, it can be assumed that a good densification was achieved in all the scaffolds for all the pore sizes. On the other hand, the heat treatment process for the evaporation of the space holder was effective. XRD analysis confirmed that ammonium bicarbonate was eliminated as no traces of this compound were found within the diffractograms.

Pore size is a crucial morphological property that regulates the osteoconductivity and osteointegration of a scaffold. Pore sizes for optimum cell proliferation and differentiation are recommended to be between 150 and 500 μm [12]. Figure 5 shows the pore size distribution analyzed by X-ray microtomography for scaffolds with pore former sizes of 250–400 μm and 400–800 μm before sintering. Three scaffolds for each pore former size were analyzed. From the analysis, it was seen that the pore size range was lower than the expected pore size, considering the pore former size. Pore size ranges from 188–438 μm (250–400 μm pore formers) and 328–765 μm (400–800 μm pore forms) were determined. This reduction in pore size by shrinkage or breakage of pore formers by compaction was already reported by [31]. In addition to shrinkage, there was the influence of the pore former shape and its influence on the evaluation. The pore formers had an elliptical and highly deformed shape as well as a rough inner surface, which can be partially divided into two pores by the evaluation algorithm. This is shown in Figure 4d.

Nonetheless, this analysis confirmed that the pore size distribution is fairly homogenous for a given pore former size and is very reproducible. Figure 5c,d shows the 3D image of the scaffolds produced. Direct data from the measurement showed that the scaffolds with pore former sizes of 250–400 μm had a surface area of 5307.51 mm$^2$ and 400–800 μm had a smaller surface area. of 3305.94 mm$^2$, including inner and outer pores. It is expected that for similar amounts of overall porosity, for smaller pore sizes, the surface area will be higher.

Figure 6 shows the relative solid density of scaffolds in x–y direction for pore former sizes of 250–400 μm and 400–800 μm. Areas in blue correspond to the pores, whereas areas in green correspond to the denser parts. It can be observed that the relative solid density was homogeneous across the cross section of both samples, which is an indication of the homogeneity of the scaffolds produced.

Figure 7 shows SEM images of cross sections of scaffolds with 250–400 μm pore former sizes coated with PLA by the infiltration method and using three different concentrations of PLA in chloroform: 3 g/100 mL, 6 g/100 mL, and 12 g/100 mL. In the images, Mg is shown in a light color, while PLA is shown in gray; although the resin is also shown in gray, a differentiation between them is possible. The coating deposited using a concentration of 3 g/100 mL was homogeneous, with an average thickness of 10 μm; but it was easily detached from the surface, as seen in Figure 7a (arrow). PLA covered the scaffold's surface, but the internal part of the pores was not homogeneously covered using this concentration. By using a concentration of 6 g/100 mL, it was possible to obtain a homogeneous external scaffold coating, with a thickness of less than 20 microns, which shows good adherence to the surface. It can also be observed that the internal scaffold's pores had a homogeneous coating and were not blocked (Figure 7b). The coating generated using a PLA with a concentration of 12 g in 100 mL of chloroform was considerably thicker than the other two coatings (average thickness 170 μm) and presented substrate adhesion problems, most likely due to its thickness. Additionally, only the pores close to the surface were coated.

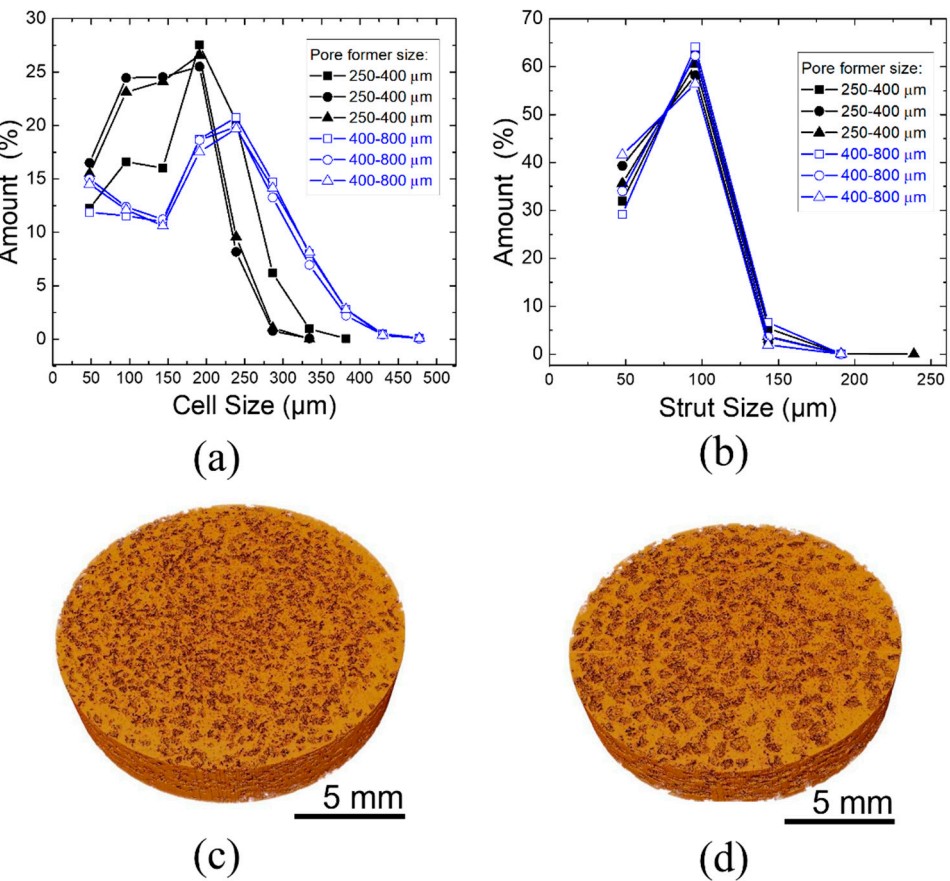

**Figure 5.** X-ray microtomography analysis of (**a**) cell size distribution and (**b**) strut thickness distribution for scaffolds produced with two sizes of pore formers: 250–400 μm and 400–800 μm. Three scaffolds for each pore size former were analyzed. The 3D images of foams with pore former sizes of (**c**) 250–400 μm and (**d**) 400–800 μm.

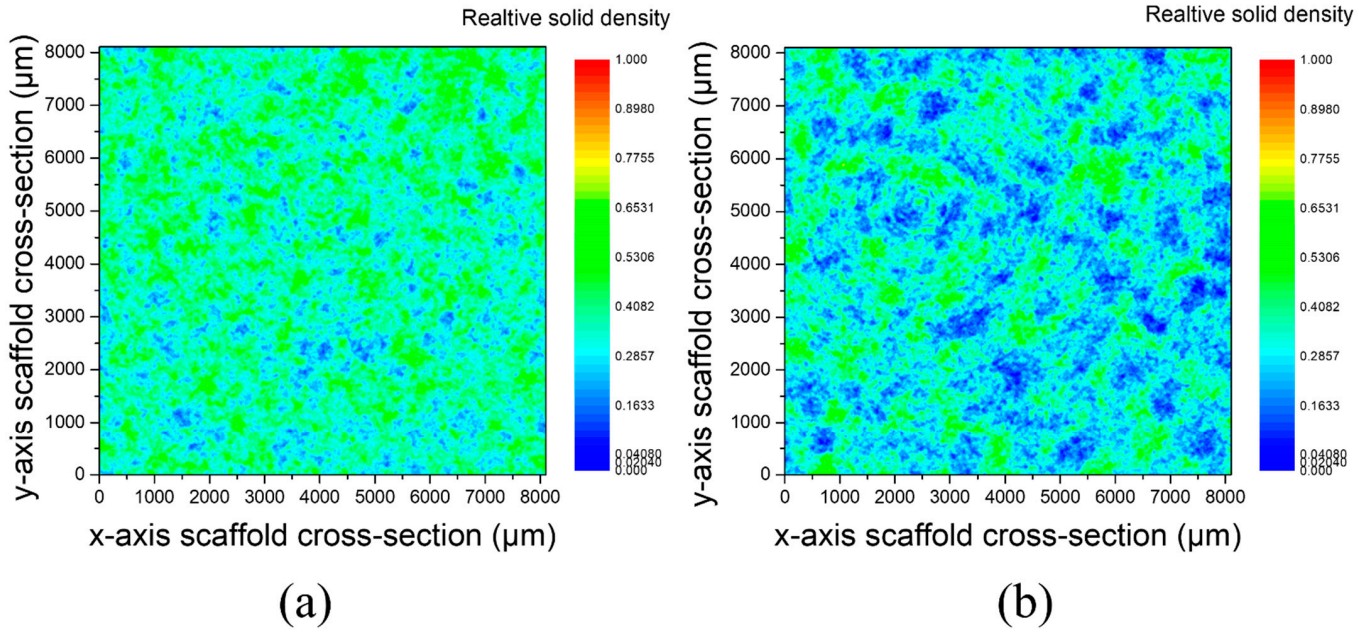

**Figure 6.** Relative solid density in x–y direction of scaffolds produced with pore former sizes of (**a**) 250–400 μm and (**b**) 400–800 μm.

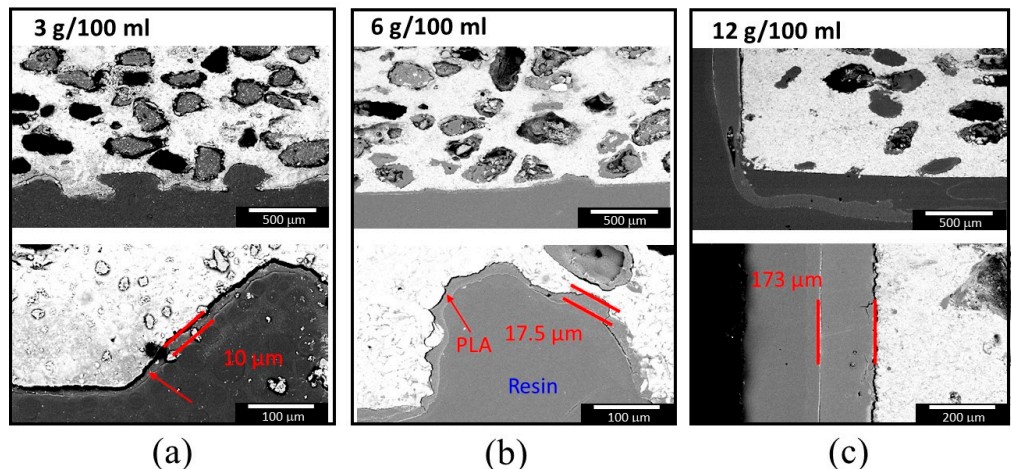

**Figure 7.** SEM images of cross sections of Mg scaffold with 250–400 μm pore former size coated with PLA using three different concentrations of PLA in chloroform: (**a**) 3 g/100 mL, (**b**) 6 g/100 mL, and (**c**) 12 g/100 mL using the vacuum-assisted infiltration technique. Upper: internal scaffold pores; Lower: external scaffold surface coating.

Figure 8 shows SEM images of cross sections of scaffolds with 400–800 μm pore size scaffolds coated with PLA using the same three different concentrations in chloroform (3 g/100 mL, 6 g/100 mL, and 12 g/100 mL). All the coatings showed good adhesion and homogeneity. The higher the concentration of PLA was , the thicker the external scaffold coating was. Namely, the external scaffold coatings obtained with solutions of 3, 6, and 12 g of PLA presented an average thickness of 10 μm, 22 μm, and 42 μm, respectively. Regarding the deposition of PLA in the interior pores by infiltration of the PLA solutions towards the interior of the samples, this was incomplete with the concentrations of 3 g/100 mL and 12 g/100 mL (Figure 8a,c). For the lowest concentration (3 g/100 mL), only a few pores in the entire sample had PLA coating. In the case of the highest concentration (12 g/100 mL), only those pores that were close to the surface had PLA coating.

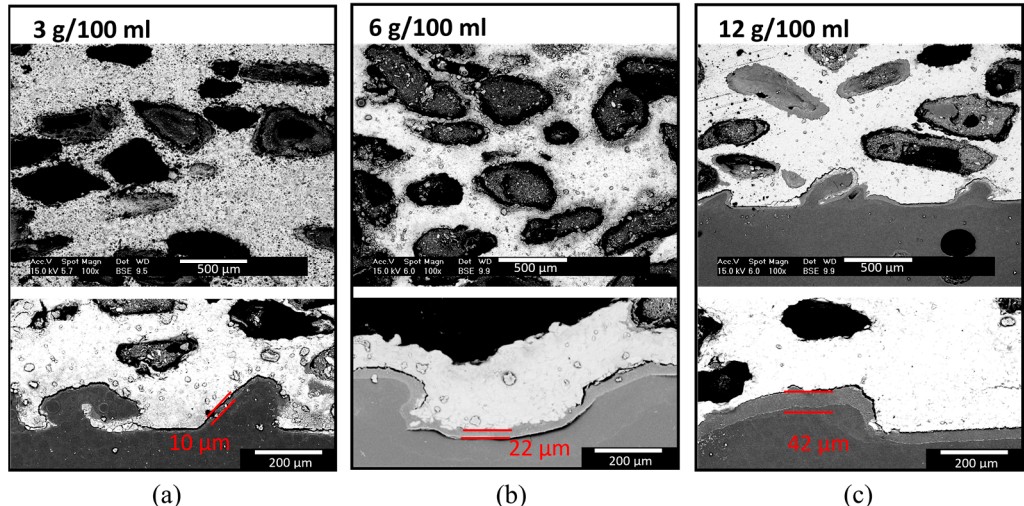

**Figure 8.** SEM images of cross sections of Mg scaffold with 400–800 μm pore size coated with PLA using three different concentrations of PLA in chloroform: (**a**) 3 g/100 mL, (**b**) 6 g/100 mL, and (**c**) 12 g/100 mL using the vacuum-assisted infiltration technique. Upper: internal scaffold pores; Lower: outer scaffold surface coating.

The difference in the coatings' thickness observable for both pore sizes can be explained taking into consideration the viscosity of the solution with the dissolved PLA and the

difference in pore size. The lower the viscosity and the larger the pore, the easier was the access of the polymer to the scaffold's interior pores. However, even though the solution with the lowest concentration was the one with the lowest viscosity, it seems that the PLA content in this solution was not sufficient to generate a coating on all the internal pores. The thickness of the coating deposited by the solution of 12 g/100 mL strongly depended on the pore size. An average thickness of 173 μm was achieved with a pore size of 250–400 μm, while a thickness of 42 μm was achieved with a pore size of 400–800 μm. This implies a 4-fold reduction in thickness. It appears that for a smaller pore size, the high viscosity of this solution impeded the PLA from reaching the internal pores and only a thick external scaffold coating layer was formed. On the other hand, for a bigger pore size, the solution can infiltrate more of the internal pores of the Mg scaffold.

The main challenge when coating a scaffold with a polymer by the deposition process is to generate a homogeneous coating with controlled thickness throughout the complete volume of the porous material. Images of cross sections of surface-treated scaffolds encountered in literature show polymeric coatings that either clogged the pores of the scaffold or coated the surface inhomogeneously. Coatings developed by Yazdimamaghani et al. [25] had an inherent porosity and they clogged the pores of Mg foam. Julmi et al. [26] coated Mg foams with PLA using a solution of 0.167 g/L in dichloromethane. Their results showed that PLA coated the surface without clogging the pores; however, the coating obtained was inhomogeneous. In the study presented here, the coating developed with a PLA concentration of 6 g/100 mL produced a homogeneous layer with a similar thickness in foams with both pore sizes. Therefore, the concentration of 6 g/100 mL was selected to continue with the study of strategies to control the in vitro degradation of magnesium scaffolds.

Figures 9 and 10 show the microstructure of the layers formed on Mg scaffolds with pore sizes of 250–400 μm and 400–800 μm, respectively, immersed for 30, 60, and 120 min in PBS solution. SEM images show that the coating formed in PBS varied in surface morphology. Three characteristic areas were identified: (1) grain-like precipitates, (2) needle-like precipitates, and (3) an area where no precipitate was found. Table 2 shows the atomic compositions of the areas identified in Figures 9 and 10 analyzed by SEM-EDX. Mg, O, P, and traces of Cl, K, and Na were found. Grain-like precipitates (1) seemed to contain similar amounts of Mg and O and, therefore, they probably corresponded to magnesium hydroxides. Needle-like areas (2) appeared to be richer in P and, therefore, these features were probably composed of phosphates. Area (3) had a higher amount of Mg and, therefore, probably corresponds to Mg that had not yet reacted. The microstructural analysis revealed that layers were mainly formed by $Mg(OH)_2$ and Mg phosphates that precipitated on the surface in a non-homogeneous way.

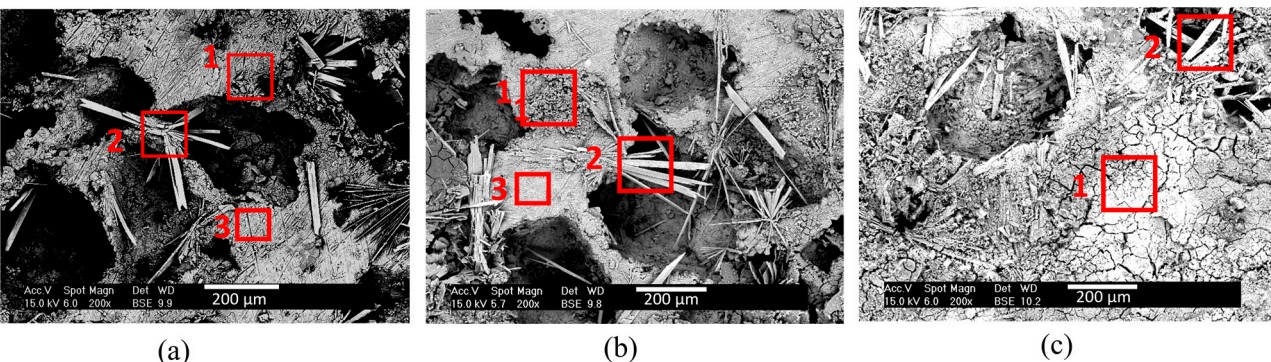

**Figure 9.** SEM microstructure of the layers formed on Mg scaffolds with pore sizes of 250–400 μm immersed in PBS solution for (**a**) 30 min, (**b**) 60 min, and (**c**) 120 min.

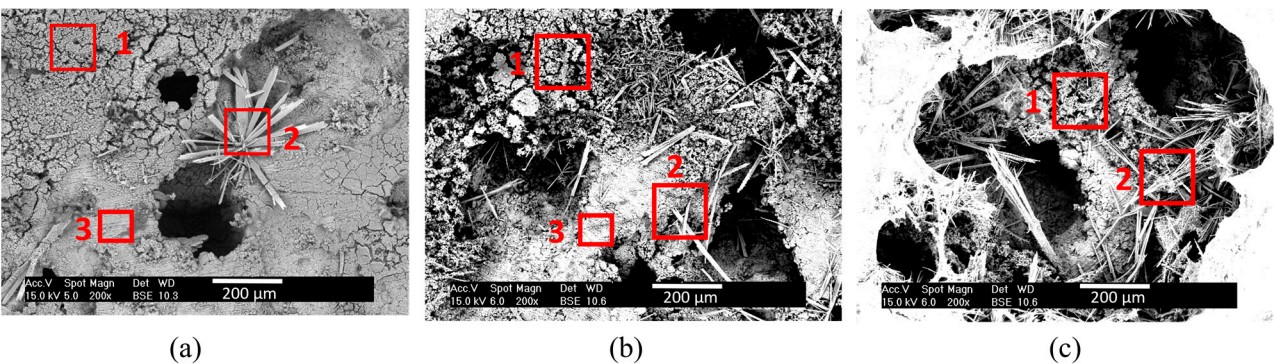

**Figure 10.** SEM microstructure of the layers formed on Mg scaffolds with pore sizes of 400–800 μm immersed in PBS solution for (**a**) 30 min, (**b**) 60 min, and (**c**) 120 min.

**Table 2.** EDX analysis of atomic compositions of areas identified in Figures 8 and 9.

| Time | Area | Pore Size 250–400 μm | | | | Pore Size 400–800 μm | | | |
|---|---|---|---|---|---|---|---|---|---|
| | | Atomic % | | | | Atomic % | | | |
| | | Mg | O | P | Cl, K, Na | Mg | O | P | Cl, K, Na |
| 30 min | 1 | 42 | 50 | 3 | 5 | 40 | 46 | 6 | 8 |
| | 2 | 17 | 55 | 16 | 12 | 20 | 47 | 17 | 16 |
| | 3 | 73 | 21 | 2 | 4 | 76 | 20 | 0 | 4 |
| 60 min | 1 | 32 | 57 | 3 | 8 | 44 | 51 | 1 | 4 |
| | 2 | 26 | 52 | 15 | 7 | 28 | 49 | 15 | 8 |
| | 3 | 75 | 22 | 0 | 3 | 78 | 17 | 1 | 4 |
| 120 min | 1 | 39 | 57 | 1 | 3 | 27 | 28 | 32 | 13 |
| | 2 | 21 | 55 | 14 | 10 | 18 | 50 | 16 | 16 |

Morphology and composition of precipitates evolves with pore size and immersion time. The bigger the pore size and the longer the immersion time, the easier is the mineralization of larger precipitates in a higher amount. Table 3 shows the Mg/O ratio and Mg/P ratio of the areas identified in Figures 9 and 10. It was found that these relationships varied with the immersion time and the pore size. Grain-like precipitates (zone 1) showed a Mg/O ratio that varied from 0.56 to 0.96 and a low content of P, with the only exception being the 400–800-μm pore size scaffold immersed for 120 min. This sample showed a Mg/O ratio closer to 1 and a high concentration of P (32% atm.). Needle-like precipitates (zone 2) showed a Mg/O ratio that varied from 0.31 to 0.57 and a higher concentration of phosphorous, showing a Mg/P ratio that went from 1.06 to 1.89. The third zone was mainly composed by magnesium and oxygen with traces of phosphorus. The Mg/O ratio here varied from 3.41 to 4.59, indicating a higher amount of Mg than oxygen. This area was not found in samples immersed for 120 min. From Tables 1 and 2 it can be inferred that, for longer times and larger pores, there was a higher amount of phosphate precipitation. The nucleation and growth of magnesium phosphate compounds depended on the scaffold surface area. Therefore, it was to be expected that, for Mg scaffolds with a higher surface area, more phosphate compounds would nucleate. Once nucleation of the phosphates occurred, the growth of the phosphates was dictated by the size of the pores: for bigger pore sizes, phosphates can grow to larger sizes. As immersion time increased, the growth of phosphates was favored and pores began to clog, particularly for longer times, as can be observed in Figure 9c. This would be detrimental for successful scaffold implementation, and, therefore, an immersion of 30 min was selected as the optimum immersion time to continue with this investigation.

**Table 3.** Mg/O ratio and Mg/P ratio of the areas identified in Figures 8 and 9.

| Time | Area | Pore Size 250–400 μm | | Pore Size 400–800 μm | |
|------|------|------|------|------|------|
| | | Mg/O | Mg/P | Mg/O | Mg/P |
| 30 min | 1 | 0.84 | 14.00 | 0.87 | 6.67 |
| | 2 | 0.31 | 1.06 | 0.43 | 1.18 |
| | 3 | 3.48 | 36.50 | 3.80 | - |
| 60 min | 1 | 0.56 | 10.67 | 0.86 | 44.00 |
| | 2 | 0.50 | 1.73 | 0.57 | 1.89 |
| | 3 | 3.41 | - | 4.59 | 78.00 |
| 120 min | 1 | 0.68 | 39.00 | 0.96 | 0.84 |
| | 2 | 0.38 | 1.50 | 0.36 | 1.13 |

The effect of the coating process on the in vitro degradation of Mg scaffolds is depicted in Figure 11, where the hydrogen release with time for scaffolds immersed in PBS solution at 37 °C was monitored. This figure shows the accumulative hydrogen release of scaffolds with pore sizes of 250–400 μm at the three conditions studied: Mg scaffold without coating (Mg), Mg scaffold coated with PLA (Mg + PLA), and Mg scaffold immersed in PBS and coated with PLA (Mg + PBS + PLA). Results are shown for Mg scaffolds coated by two different methods (dip-coating and infiltration process). It is evident from Figure 11 that the coating process had a huge influence on the degradation behavior of porous Mg. Scaffolds treated in PBS and coated with PLA by the infiltration process released the lowest amount of hydrogen. In contrast, Mg + PBS + PLA scaffolds coated by dip-coating released the highest amount of gas, even higher than uncoated Mg scaffolds. During the first 30 h, all coated scaffolds released a lower amount of hydrogen than the uncoated Mg scaffold. However, from the third day on, the behavior of the dip-coated samples was similar or worse than the uncoated samples. These results demonstrated that the deposition methodology of the PLA layer strongly influenced the quality of the coating. The vacuum infiltration process generated homogeneous coatings throughout all the scaffold volume that achieved a good control of Mg corrosion with time.

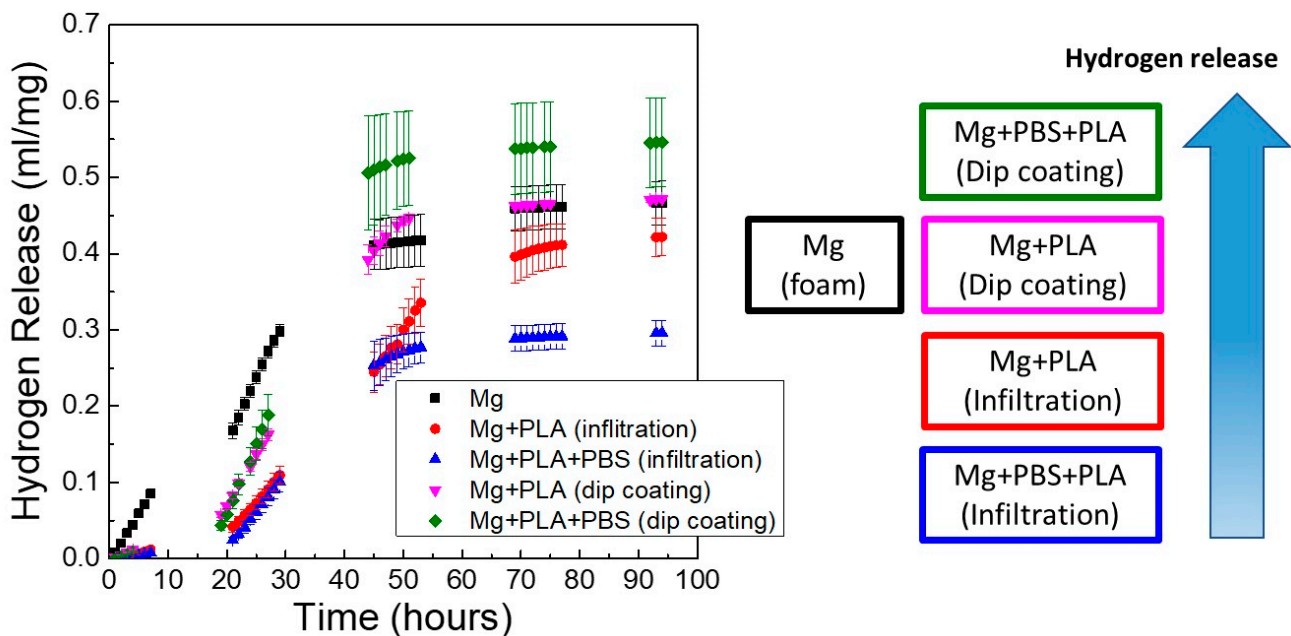

**Figure 11.** Effect of coating process on the in vitro degradation (hydrogen release) of Mg scaffolds immersed in PBS solution at 37 °C for scaffolds with pore sizes of 250–400 μm.

The effect of pore size on the in vitro degradation kinetics of scaffolds immersed in PBS solution is shown in Figure 12. The depicted data correspond to those scaffolds coated using the infiltration method. Scaffolds with pore sizes of 250–400 μm and 400–800 μm were compared. Scaffolds with pore sizes 400–800 μm had a slightly higher degradation rate than scaffolds with pore sizes of 250–400 μm. Similar results were found by M.-Q. Cheng et al. [19], who compared the corrosion rate of scaffolds with the same porosity and different pore size and found a slightly higher corrosion rate for scaffolds with larger pore size. In Figure 12 it is also observed that PLA and bilayer coatings can control the degradation behavior of Mg scaffolds. Interestingly, the bilayer coating (PBS + PLA) was more effective at controlling the rate of hydrogen release in the 250–400-μm pore size scaffolds than in the scaffolds with a larger pore size (400–800 μm). Regarding scaffolds with the largest pore size, the uncoated scaffolds exhibited rapid in vitro degradation, followed by the scaffold with the PBS + PLA treatment; the one that degraded in the most controlled way was the one coated only with PLA. The effectiveness of the bilayer coating depended on the pore size, being more effective for the pore size of 250–400 μm than for the pore size of 400–800 μm. This could be explained given the different morphologies obtained of the conversion layer for each pore size. Figure 9a shows the morphology of the layer that precipitated on the scaffold with the smaller pore size. In that image, the pores of the scaffold are still visible and their size had not been reduced by the precipitates, whereas in Figure 10a, it is observed that the pores of 400–800 μm, although they were not completely clogged, had larger phosphate and hydroxide precipitates on the surface. The nucleation and growth of precipitates clearly influenced and limited the wetting behavior of the PLA solution, leading to a less homogeneous polymeric coating and a poorer control over the hydrogen release for the scaffolds with bigger pore size.

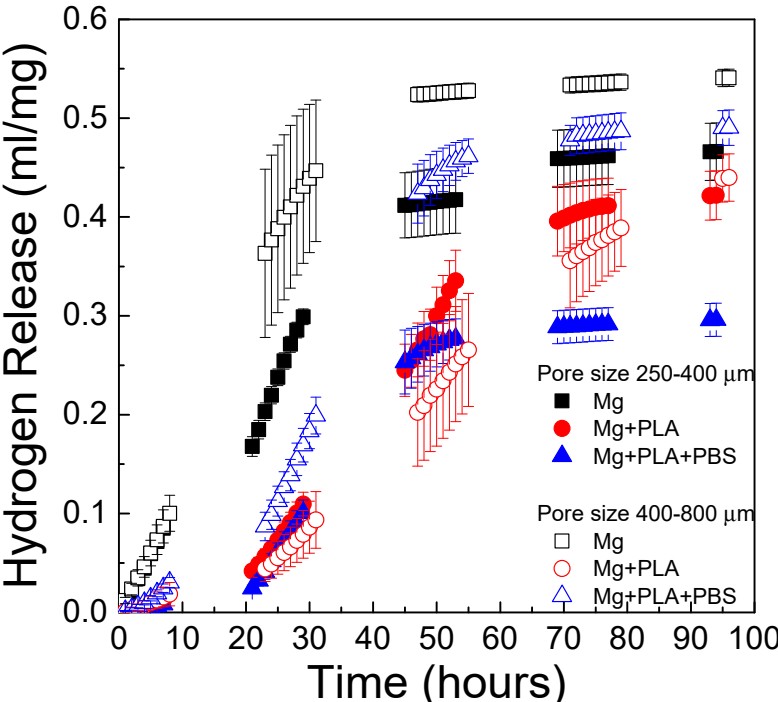

**Figure 12.** Effect of pore size on the in vitro degradation (hydrogen release) of Mg scaffolds immersed in PBS solution at 37 °C for scaffolds with the infiltration method.

The pH evolution with immersion time of the PBS solution containing the scaffolds is shown for each pore size in Figure 13. While uncoated Mg increased the pH of the PBS solution over 8 in 40 min, coated samples with PLA and PBS + PLA increased the pH of PBS to 7.5 after 2 h. Therefore, the coatings succeeded at controlling the pH of a simulated body fluid for short times.

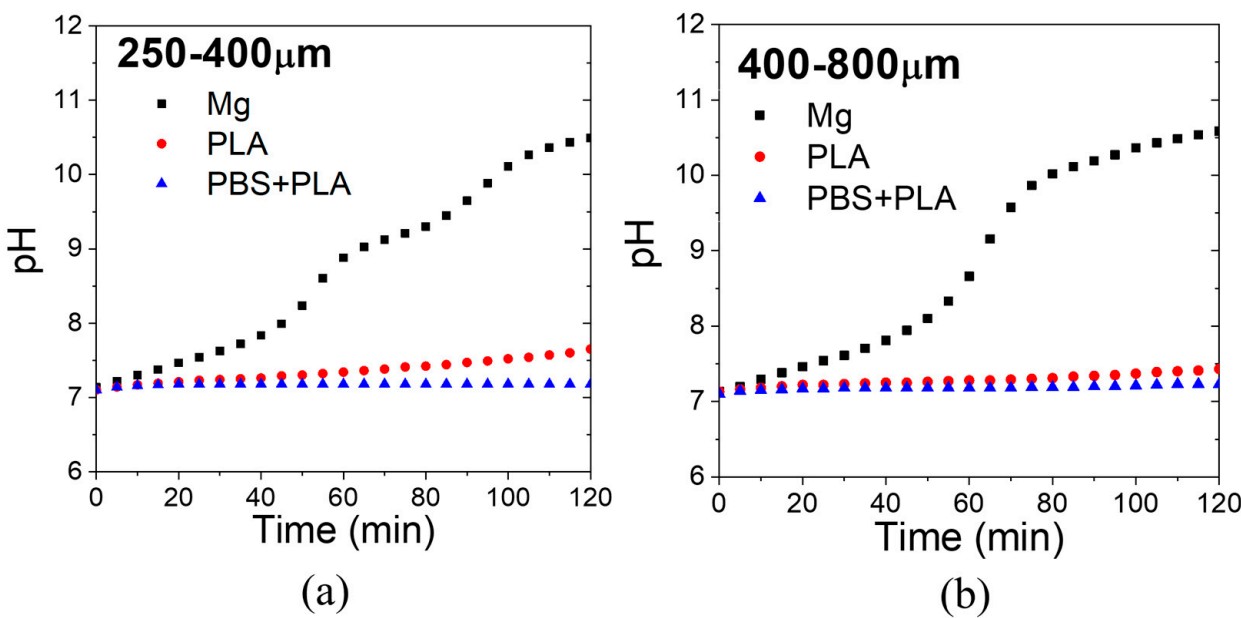

**Figure 13.** The pH evolution of PBS solution with scaffolds with pore sizes of (**a**) 250–400 μm and (**b**) 400–800 μm.

### 4. Conclusions

Powder metallurgy Mg scaffolds were produced by a simple, reproducible method: using space holders, which allowed successful control of pore and strut size. A novel bilayer coating based on PLA and co-precipitated protective hydroxides and phosphates was applied to the Mg scaffolds in order to control their degradation kinetics in vitro. A homogenous coating was applied, which did not clog the scaffold's cells. This was achieved using a vacuum impregnation technique, which ensured complete deposition of the scaffold's cells with the protective coating.

Scaffold pore size influenced both PLA deposition and hydroxide and phosphates' precipitation. Bigger pore size favored a larger amount of hydroxide and phosphate precipitation. Regarding PLA deposition, for lower concentrations, pore size did not seem to affect the capacity impregnation and thickness of the PLA layer deposited. However, for increasing PLA concentrations, only scaffolds with a bigger pore size were successfully impregnated. The scaffolds with the protective coating produced by the vacuum impregnation technique showed a more controlled degradation in vitro as compared to Mg uncoated scaffold and scaffolds coated using dip-coating. For smaller pore sizes, the scaffolds benefited from the bilayer coating, as compared to a single PLA coating. However, for a bigger pore size, the excessive amount and size of hydroxide and phosphates' precipitates were detrimental for the homogeneity of the bilayer coating, and single PLA coating exhibited better behavior. Therefore, for optimum control of the degradation of magnesium scaffolds, it is important to consider the effect of pore size on the precipitation of the chemical conversion layer as well as ensuring homogeneous PLA deposition. Finally, it was proved that the bilayer coating containing hydroxides and phosphates was successful in controlling Mg scaffold degradation, as long as the size and amount of the hydroxides' and phosphates' precipitates were controlled and the PLA coating remained homogenous.

**Author Contributions:** The conceptualization was designed by S.C.C. The methodology was designed and improved by S.C.C. and S.A.T. The formal analysis, investigation, and validation were carried out by all authors: S.C.C., S.A.T., T.F., L.A. (Lucía Alvarez) and L.A. (Luis Arias). The writing of the original draft was prepared by S.C.C. The reviews and editing of the manuscript were made by S.C.C., S.A.T. and T.F. The supervision was made by S.C.C. and S.A.T. All authors have read and agreed to the published version of the manuscript.

**Funding:** Funding was provided by the Regional Government of Madrid through the project P2018/NMT4411 (ADITIMAT-CM) and the Spanish Government through the projects PID2019-106631GB-C43 and RTC2019-007049-4.

**Institutional Review Board Statement:** Not applicable.

**Informed Consent Statement:** Not applicable.

**Data Availability Statement:** Not applicable.

**Acknowledgments:** The authors would like to thank the funding provided for this research by the Regional Government of Madrid (Dra. Gral. Universidades e Investigación) through the project P2018/NMT4411 (ADITIMAT-CM) and the Spanish Government through the projects PID2019-106631GB-C43 and RTC2019-007049-4.

**Conflicts of Interest:** The authors declare no conflict of interest.

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
