# Peer review of "Strategies to Control In Vitro Degradation of Mg Scaffolds Processed by Powder Metallurgy"

_metals, doi:10.3390/met12040566_

Round 1
Reviewer 1 Report
Porous Mg scaffolds and Mg scaffolds modified by PBS and PLA were fabricated in this manuscript. The pore structures and biodegradable properties of the specimens were characterized in detail. The manuscript can be published after revising several deficiencies.
- Is the method of preparing coating described in Figure 2 dip coating? The authors are encouraged to provide the schematic diagram of vacuum-assisted infiltration method.
- The way to express milliliters should be mL.
- The particle size of space holders shown in Figure 4a is not “>125 μm”.
- Why didn’t the authors provide SEM images of the Mg scaffolds with PBS and PLA coating?
Author Response
Dear Reviewer, we have carefully read your comments and have answered and made modifications as suggested. We would like to thank you for the time spent considering our manuscript for publication in Metals and your valuable suggestions made.
We have listed changes below, and have highlighted the changes in yellow in the manuscript.
Porous Mg scaffolds and Mg scaffolds modified by PBS and PLA were fabricated in this manuscript. The pore structures and biodegradable properties of the specimens were characterized in detail. The manuscript can be published after revising several deficiencies.
- Is the method of preparing coating described in Figure 2 dip coating? The authors are encouraged to provide the schematic diagram of vacuum-assisted infiltration method.
Response: The reviewer is right. We have added a schematic diagram in Figure 2 describing both, dip coating and the vacuum assisted infiltration method.
2. The way to express milliliters should be mL.
Response: We have replaced ml by mL as requested by reviewer.
3. The particle size of space holders shown in Figure 4a is not “>125 μm”.
Response: Thank you so much for addressing this mistake. You are completely right it is not > 125 μm, it is < 125 μm. We have changed that in Figure 4a.
4. Why didn’t the authors provide SEM images of the Mg scaffolds with PBS and PLA coating?
Response: The fragile nature of the precipitates generated after immersion in PBS fluid make metallographic observation not ideal for observing and identifying the morphology and composition of these precipitates. Authors considered that the best way to identify the composition and morphology of the precipitates formed after immersion in PBS would be to observe the scaffold surface itself, rather than subject it to aggressive metallographic preparation procedures that might alter its characteristics. The preparation of cross sections and surfaces of Mg scaffolds with PBS and PLA damaged the coatings and altered their characteristics.
Reviewer 2 Report
The presented investigation deals with biodegradable materials based on the magnesium scaffolds for orthopedics needs. The main highlights of this paper focus on the study of porosity of the magnesium scaffolds, which influence the cell proliferation for future tissue growth in vivo. It was proposed a novel approach by the way of obtaining a double-layer coating on the surface of the scaffold which should lead to a decrease of a high rate of magnesium corrosion. This work is aimed at solving a concrete problem and therefore it is scientifically significant. The paper is well organized and the investigation includes modern methods such as a micro-KT, that have increased the accuracy of the obtained results on the level of the depth of studying of the problem. However, there are some remarks and questions.
From the presented introduction it still does not clear which pore size is more perfect for magnesium scaffolds. It will be interesting to mention the influence of the open and close porosity on the process of cells proliferation.
From the paragraph about scaffolds characterization, it does not understand what is a “theoretical porosity”, and why in figure 3 porosity is characterized by the value of 40%? The authors should connect open and close porosity with a theoretical density of the scaffolds by the determination of geometry density in this case.
Should check the subscript on figure 3 (the absent a and b symbols)
From XRD data of the sintered scaffolds, it will be interesting to observe the Quantitative value of magnesium oxide in the structure, and there is a difference between the atmosphere of sintering?
Figure 6 check the subscript.
From the point of view of in vitro studies, it would be more accurate to perform the long-time experiment (for example, 1,3,7,14 and 21 days) for evaluating the dynamics of magnesium scaffold corrosion indicators (pH level, hydrogen evolution, and weight loss).
Author Response
Dear Reviewer, we have carefully read your comments and have answered and make modifications as suggested. We would like to thank you for the time spent considering our manuscript for publication in Metals and your valuable suggestions made.
We have listed changes below, and have highlighted the changes in yellow in the manuscript.
The presented investigation deals with biodegradable materials based on the magnesium scaffolds for orthopedics needs. The main highlights of this paper focus on the study of porosity of the magnesium scaffolds, which influence the cell proliferation for future tissue growth in vivo. It was proposed a novel approach by the way of obtaining a double-layer coating on the surface of the scaffold which should lead to a decrease of a high rate of magnesium corrosion. This work is aimed at solving a concrete problem and therefore it is scientifically significant. The paper is well organized and the investigation includes modern methods such as a micro-KT, that have increased the accuracy of the obtained results on the level of the depth of studying of the problem. However, there are some remarks and questions.
From the presented introduction it still does not clear which pore size is more perfect for magnesium scaffolds. It will be interesting to mention the influence of the open and close porosity on the process of cells proliferation.
Response: Thank you so much for your comment. We agree that it is important to mention the influence of open and close porosity on cells proliferation. We have included this information in the introduction and added references [11] and [12].
Scaffolds’ porosity is a crucial morphological property to enhance cell proliferation and tissue ingrowth. Interconnected porous networks are preferred rather than closed pores. Open porosity is osteoconductive, it allows cell migration and vascularization [11]. Pore sizes for optimum cell proliferation and differentiation are recommended to be between 150 and 500 μm [12].
[11] G. Ryan, A. Pandit, D.P. Apatsidis, Fabrication methods of porous metals for use in orthopaedic applications, Biomaterials. 27 (2006) 2651-2670. https://doi.org/10.1016/j.biomaterials.2005.12.002
[12] V.M. Goldberg, S. Akhavan, Biology of Bone Grafts BT - Bone Regeneration and Repair: Biology and Clinical Applications, in: J.R. Lieberman, G.E. Friedlaender (Eds.), Humana Press, Totowa, NJ, 2005: pp. 57–65. https://doi.org/10.1385/1-59259-863-3:057.
From the paragraph about scaffolds characterization, it does not understand what is a “theoretical porosity”, and why in figure 3 porosity is characterized by the value of 40%? The authors should connect open and close porosity with a theoretical density of the scaffolds by the determination of geometry density in this case.
Response: Theoretical porosity refers to the porosity that the scaffold would present if all porosity was solely due to the addition of space holder. That is, if 40 % of space holder was added to Mg powder during scaffold preparation, the theoretical porosity of this scaffolds should be 40%. Modifications have been made to the text to clarify this point.
Should check the subscript on figure 3 (the absent a and b symbols)
Response: Thank you so much for addressing this mistake. You are completely right, symbols a) and b) are absent, we have added them in Figure 3.
From XRD data of the sintered scaffolds, it will be interesting to observe the Quantitative value of magnesium oxide in the structure, and there is a difference between the atmosphere of sintering?
Response: Due to the low relative intensity of the magnesium oxide peak, and the fact that only the main peak of magnesium oxide is present, with secondary peaks not visible, it is very difficult to perform an accurate quantification, and the errors in doing so would be big. Comparing the relative intensities of the MgO peaks for both atmospheres, it could be deduced that the amount of Mg oxide formed is very similar for both atmospheres. Considering that only the main Mg oxide peak can be clearly identified in the XRD spectra, it is possible that the amount of MgO present is close to the detection limit of the technique, which is 5%.
Figure 6 check the subscript.
Response: thank you very much for you comment. The subscript in figure 6 has been checked
From the point of view of in vitro studies, it would be more accurate to perform the long-time experiment (for example, 1,3,7,14 and 21 days) for evaluating the dynamics of magnesium scaffold corrosion indicators (pH level, hydrogen evolution, and weight loss).
Response: Thank you very much for your comment. We agree and aim to continue the study and perform long-time experiments. However, at present, we feel that this is a good initial investigation for a promising way to control in-vitro degradation of Mg scaffolds
Reviewer 3 Report
Please check the theoretical porosity mark in Figure 3.
It is suggested to beautify the charts.
The high activity of magnesium makes it very easy to oxidize. The authors mixed magnesium powder with ammonium bicarbonate to prepare porous magnesium scaffolds, and the size of the pores could be adjusted. This method is innovative. Meanwhile, a novel bi-layer coating, based on PLA and co-precipitated protective hydroxides and phosphates was applied to the Mg scaffolds in order to control their degradation kinetics in-vitro. This research can provide some help for later researchers and help to promote the development of magnesium biomaterials.
For the above reasons, I think this manuscript has a certain novelty and scientificity.
Author Response
Dear Reviewer, we have carefully read your comments and have answered and make modifications as suggested. We would like to thank you for the time spent considering our manuscript for publication in Metals and your valuable suggestions made.
We have listed changes below, and have highlighted the changes in yellow in the manuscript.
Please check the theoretical porosity mark in Figure 3.
Response: Thank you very much. The mark has been modified
It is suggested to beautify the charts.
Response: Thank you very much. All charts have been beautified according to the format suggested by METALS.
The high activity of magnesium makes it very easy to oxidize. The authors mixed magnesium powder with ammonium bicarbonate to prepare porous magnesium scaffolds, and the size of the pores could be adjusted. This method is innovative. Meanwhile, a novel bi-layer coating, based on PLA and co-precipitated protective hydroxides and phosphates was applied to the Mg scaffolds in order to control their degradation kinetics in-vitro. This research can provide some help for later researchers and help to promote the development of magnesium biomaterials.
For the above reasons, I think this manuscript has a certain novelty and scientificity.
Response: Thank you very much for your kind review
Round 2
Reviewer 2 Report
The paper was improved according to recommendations.
Author Response
Thank you for your comments.